# The Discrimination and Characterization of Volatile Organic Compounds in Different Areas of *Zanthoxylum bungeanum* Pericarps and Leaves by HS-GC-IMS and HS-SPME-GC-MS

**DOI:** 10.3390/foods11223745

**Published:** 2022-11-21

**Authors:** Xinlong Wu, Jiaxin Yin, Hui Ding, Wei Li, Lifeng Han, Wenzhi Yang, Fangyi Li, Xinbo Song, Songtao Bie, Xingchu Gong, Heshui Yu, Zheng Li

**Affiliations:** 1College of Pharmaceutical Engineering of Traditional Chinese Medicine, Tianjin University of Traditional Chinese Medicine, Tianjin 301617, China; 2State Key Laboratory of Component-Based Chinese Medicine, Tianjin University of Traditional Chinese Medicine, Tianjin 301617, China; 3Haihe Laboratory of Modern Chinese Medicine, Tianjin University of Traditional Chinese Medicine, Tianjin 301617, China; 4Pharmaceutical Informatics Institute, College of Pharmaceutical Sciences, Zhejiang University, Hangzhou 310058, China

**Keywords:** headspace gas chromatography-ion mobility spectrometry, headspace solid phase microextraction-gas chromatography-mass spectrometry, *Zanthoxylum bungeanum*, fingerprint, principal component analysis, orthogonal partial least squares discriminant analysis

## Abstract

The pericarps of *Zanthoxylum bungeanum* (ZBP) and leaves of *Zanthoxylum bungeanum* (ZBL) are popular spices in China, and they have pharmacological activities as well. In this experiment, the volatile organic compounds (VOCs) of the pericarps of *Zanthoxylum bungeanum* in Sichuan (SJ) and its leaves (SJY) and the pericarps of *Zanthoxylum bungeanum* in Shaanxi (SHJ) and its leaves (SHJY) were analyzed by headspace-gas chromatography-ion mobility spectrometry (HS-GC-IMS) and headspace solid phase microextraction-gas chromatography-mass spectrometry (HS-SPME-GC-MS). The fingerprint of HS-GC-IMS and the heat maps of HS-SPME-GC-MS were established. Principal component analysis (PCA) and orthogonal partial least squares discriminant analysis (OPLS-DA) were performed. The results showed that a total of 95 components were identified, 62 components identified by HS-SPME-GC-MS and 40 components identified by HS-GC-IMS, of which 7 were the same. The analysis found that SJ and SHJ were obviously distinguished, while SJY and SHJY were not. There were considerably fewer VOCs in the leaves than in the pericarps. In the characterization of the VOCs of ZBL and ZBP, the flavor of ZBP was more acrid and stronger, while the flavor of ZBL was lighter and slightly acrid. Thirteen and eleven differential markers were identified by HS-GC-IMS and HS-SPME-GC-MS, respectively. This is helpful in distinguishing between SHJ and SJ, which contributes to their quality evaluation.

## 1. Introduction

The pericarps of *Zanthoxylum bungeanum* (ZBP) are an important seasoning in China [1]. ZBP has a special flavor and is often used in Sichuan dishes. In addition to that, it is in great demand in traditional medicine [2]. Essential oils are its main active ingredients. It has anti-inflammatory [3], antiseptic [4] and other effects. *Zanthoxylum bungeanum* is grown in southwest China in Sichuan, Shaanxi, Yunnan and so on [5]. Its wide range of cultivation leads to different quality. It is necessary to differentiate ZBP from different regions. Aroma has an important influence on food evaluation results [6], and the volatile terpenoids are the main source of its aroma [7]. People have different opinions about ZBP from different producing areas: its color, taste, smell and even its medicinal effects. For the evaluation of the quality of pepper, experienced operators are often required. However, there is also subjectivity in judgment that affects the stability and accuracy of quality evaluation. Therefore, it is necessary to find a suitable approach to identifying it. Other than that, ZBL has special aroma. It is also used as a condiment and has some antioxidant [8] and antibacterial activity [9]. Currently, the study of volatile organic compounds (VOCs) of ZBP is abundant, but the study of VOCs of Leaves of *Zanthoxylum bungeanum* (ZBL) is less. It is interesting to characterize the VOCs of ZBL and investigate the similarities and differences of their composition with ZBP.

GC-MS is undoubtedly a good choice for the detection of VOCs, high sensitivity, accuracy and wide detection range, but it is a complex treatment of samples [10]. Solid-phase microextraction (SPME) has the advantages of good extraction effect, selectivity, environmental friendliness and convenience [11]. The SPME combination with GC-MS can make up for some of its shortcomings, mainly the complex preprocessing steps. It was reported that SPME-GC-MS combined with chemometrics could be used to analyze the quality of strong aroma base liquor at different grades [12].

Ion mobility spectrometry (IMS), as a high-sensitivity, fast detection method, was used in the military early [13]. GC has good separation performance. By coupling GC with IMS, HS-GC-IMS ushered in a new development in food [14], cosmetics [15] and medicine [16]. In HS-GC-IMS, VOCs form into ions when they pass through the ionization source, and the ions drift in a weak electric field at atmospheric pressure; depending on the differences in structure, mass, charge and volume of each ion, the ions are detected at different times, providing information on the type and concentration of the analyte [17]. In addition, HS-GC-IMS also has the advantages of high sensitivity and environmental friendliness and no sample pretreatment [18]. Previously, it has been reported that HS-GC-IMS can identify Pericarpium Citri Reticulatae and its counterfeits [19]. GC-MS tends to be qualitative and quantitative. HS-GC-IMS tends to identify samples and can also detect small odor molecules that are not detected by GC-MS. GC-MS and HS-GC-IMS each have their own advantages. The combination of these two approaches can achieve better evaluation results [20]. It has reported that the Liuyang Douchi was determined by HS-GC-IMS and HS-SPME-GC-MS [21].

Maoxian and Hancheng are historical pepper production areas in China. Maoxian pepper has been appraised as one of China’s national geographical indication products. The purpose of this study was to characterize the VOCs of ZBP of Hancheng (SHJ), ZBL of Hancheng (SHJY), ZBP of Maoxian (SJ) and ZBL of Maoxian (SJY) based on HS-GC-IMS and HS-SPME-GC-MS and investigate their differential components. Laying the foundation for the study of its active ingredients, the fingerprint of HS-GC-IMS data and the heat map of HS-SPME-GC-MS data were established. The PCA and OPLS-DA were used to distinguish the samples and find the differences in VOCs between them. Differential marker screening of ZBP and ZBL by one-way ANOVA and variable importance in projection (VIP) evaluation for the differential marker were calculated.

## 2. Materials and Methods

### 2.1. Materials and Chemicals

In this study, ZBP and ZBL taken from Maoxian, Sichuan, China (longitude 102.22 east latitude 31.90 north), and Hancheng, Shaanxi, China (east longitude 110.43 north latitude 35.48) (Appendix A) were selected (for detailed information, refer to Appendix A). Six batches were selected from each producing area to pick leaves and fruits in July. The N-ketone C4-C9 standard mix purchased from Sinopharm Chemical Co., Ltd. (Shanghai, China) was used to calculate retention index (RI) of HS-GC-IMS, N-alkane C8-C20 standard mix for GC–MS was purchased from Sigma-Aldrich Chemical Co., Ltd. (Saint Louis, MO, USA). The standard compounds are as follows: linalool (CAS registry No. 78-70-6, purity ≥ 98%), linalyl acetate (CAS registry No. 115-95-7, purity ≥ 98%), l-menthol (CAS registry No. 2216-51-5, purity ≥ 98%), myrcene (CAS registry No. 123-35-3, purity ≥ 98%,) all purchased from Nature Standard (Shanghai, China).

### 2.2. HS-GC-IMS Analysis

The GC-IMS system (FlavourSpec^®^, Gesellschaft für Analytische Sensorsysteme mbH, Dortmund, Germany) with an autosampler (CTC Analytics AG, Zwingen, Switzerland) was used to detect the VOCs of that ZBP and ZBL in Sichuan and Shaanxi. Weigh three samples from each batch for experiment. Weigh 0.01 g of each batch of samples and add them into a 20 mL headspace glass sampling bottle (Zhejiang HAMAG technology, Ningbo, China). Set the relevant parameters of the instrument as follows.

Conditions of HS-GC-IMS: incubation temperature was 75 °C, the incubation time was 20 min, the oscillation rate was 500 rpm, headspace injection was used, the injection temperature was 85 °C, the injection volume was 300 µL, and the carrier gas was N2 (purity ≥ 99.999%). The column was FS-SE-54-CB-1 capillary column (15 m × 0.53 mm ID, 1 µm, Beijing, China), the column temperature was 60 °C, the programmed flow was as follows: 2 mL/min initially for 2 min, increased to 10 mL/min over 10 min, increased to 150 mL/min over 20 min, and keep the flow rate of 150 mL/min over 30 min. The drift tube is 98 mm long, the temperature is 45 °C, the voltage is 500 kV, and the drift gas is nitrogen (purity ≥ 99.999%). The gas flow of drift tube was 150 mL/min.

### 2.3. SPME Optimization

SPME fiber (Supelco, Bellefonte, Penn.) was installed on a MultiPurpose sampler (Gerstel, GER) and combined with 7890B-7000D triple quadrupole gas chromatography mass spectrometry (Agilent Technologies, Palo Alto, CA, USA) to detect VOCs that ZBP and ZBL. Weigh three samples from each batch for experiment. weigh 0.01g for 20 mL headspace vials for each batch of samples. Incubation temperature was 75 °C, incubation time was 5 min, extraction time was 40 min, vial penetration was 21.00 mm, injection penetration was 54.00 mm desorption time was 5 min, desorption temperature was 250 °C.

### 2.4. Conditions of GC-MS

The chromatographic column was HP-5MS phenyl methyl siloxane (30 m × 0.25 mm × 0.25 µm, Agilent Technologies, Palo Alto, CA, USA) elastic quartz capillary column; The initial temperature was 50 °C for 2 min, then 20 °C/min to 80 °C, and 2 °C/min to 155 °C, finally 30 °C/min to 220 °C and held for 5 min; The carrier gas was Helium (purity 99.999%) with a flow rate of 1.0 mL/min; The split ratio was 1:1; The injection port temperature is 250 °C. The conditions of MS were set as follow: the ion source temperature was 230 °C, ionization energy was 70 eV with a scan range of *m*/*z* 30ߝ650.

### 2.5. Statistical Analysis

Weighed an equal amount of sample as QC, each take 6 QC measured by HS-GC-IMS and HS-SPME-GC-MS, and selected 9 and 6 major components to calculate the relative standard deviation to verify the reproducibility, RSD < 10.47% and 6.67%, respectively (Appendix A). Analysis of HS-GC-IMS data based on Vocal, drift time and RI were used as criteria for component identification. Analysis of GC-MS data based on Masshunter. NIST17 (match > 80, RI) as well as standard compounds were used for identification analysis. Multivariate statistical analysis by simca 14.1 (Umetrics, Malmo, Sweden). Data were normalized by peak area and normalization and scaled using pareto scaling. Plotting heat maps on MetaboAnalyst 5.0 (https://www.metaboanalyst.ca/, accessed on 11 June 2022). Compound odor and partial activity query from ChemicalBook (www.chemicalbook.com, accessed on 15 June 2022).

## 3. Results and Discussion

### 3.1. HS-GC-IMS Topographic Plots in SHJY, SHJ, SJY and SJ

In HS-GC-IMS, when the concentration of components in the sample is high, the compounds will form dimers or even multimers with protons or electrons. And the presence of multimers contributes more to qualitative accuracy. The X-axis was the drift time, the Y-axis was the retention time, and the Z-axis was the ion peak intensity. The color represented the peak intensity. The peak intensity decreased from red to white and then to blue. It can be seen that most of the ion peaks appear at the retention time of 100–1600 s and the drift time of 1.0–2.5 (Figure 1A). Identification of differences in SHJY, SHJ, SJY, SJ from three dimensions of retention time, drift time and peak intensity. The overall number of peaks of ZBP was more than that of ZBL, and the peak intensity was greater than that of ZBL. Compared with the other three samples, SHJ had two very obvious ion peaks. There was little difference in the peaks of the same part from different places, but the difference in the peak can also be seen.

### 3.2. Fingerprint Analysis of VOCs in SHJ, SJ, SHJY, SJY

The aroma is usually composed of small molecule volatile aldehydes, ketones, organic acids, esters and monoterpenes [22]. There were 5 aldehydes, 5 olefins, 10 alcohols, 5 esters, 4 acids, 6 ketones, 1 ether, 3 thioethers and 1 phenol detected by HS-GC-IMS (Table 1). Figure 1B shows the fingerprint of SHJ, SJ, SHJY, and SJY. The peak intensity decreased from red to white, and then to blue. It can be seen that there are more components in ZBL in Figure 1B-a. E-2-hexenal (strong aroma of fresh fruit and green leaves), diethyl butanedioate (pleasant smell), ethyl acetate (strong etherlike smell, clear and fruity aroma). The components of the pericarps and its leaves in Figure 1B-b were both obvious. Such as pentanal (special fragrance), *α*-terpinene (citrus and lemonlike aroma, antibacterial and antioxidant), ethanol, 1,8-cineole (camphor smell and cool herbal flavor), *β*-pinene (resin and rosin aroma), *α*-pinene (anti tumor, antifungal, anti allergy and ulcer improvement). The components of the pericarps in area Figure 1B-c were more significant, Such as 2-hexanol, acetic acid, 4-terpineol (aroma of pepper, light earth and stale wood), furaneol (sweet, toasty, bread, cooking, fruit and caramel frangrance), *δ*-octalactone (cocoa, coconut and creamy aromas). These ingredients are also a little in ZPL. Methyl-5-hepten-2-one (fruit aroma and fresh fragrance), (E,E)-2,4-hexadienal, furfural (special smell similar to Benzaldehyde), (E)-2-pentenal, neryl acetate (aromas of orange blossom and rose, honey and raspberry), butanoic acid, nerol (sweet smell of fresh roses, with a slight smell of lemon), 2-butoxyethanol (moderate ether flavor), phenylacetic acid, 2,3-butanedione (creamy aroma after dilution), acetone, allyldisulfide (special smell of garlic), carveol. Nerol has antioxidant, antimicrobial, antispasmodic, insect repellent and antiarrhythmic effects. A certain dose of nerol could protect the liver of rats [23]. 4-terpineol had strong antifungal activity and was one of the main antifungal components of tea tree oil [24]. There were more components of SJ in Figure 1B-d, such as linalool oxide, isopropyl acetate (sruity), 2-Cyclohexen-1-one, 2,3-Butanediol, and *β*-ocimene was also obvious in ZBL. The components of SHJ pericarps in Figure 1B-e were relatively high. Such as pentanol, and it has a certain content in SHJY, SJY and SJ, 2,5-dimethylfuran, dimethyl disulfide, 2-methyl butanol, 2,3-Butanediol, 2-acetylfuran (slmonds, nuts, fermented, milk and sweet caramel-like aroma), dipropyl sulfide, phenylacetic acid, 4-methylguaiacol (aroma of spices, cloves, vanillin and smoke).

### 3.3. Component Content Analysis of HS-SPME-GC-MS Results

The overall peak intensity of ZBP was higher than ZBL, the overall peak intensity of SJ was higher than that of SHJ, and SHJ has more peak than SJ. The top three VOCs in the peak area of each sample were as follows: linalool 25.54%, linalyl acetate 18.56%, and terpinyl acetate 4.89% of SJY; linalyl acetate 21.26%, linalool 13.6%, terpinyl acetate 12.16% of SHJY; Dl-limonene 3.99%, piperitone 3.67% and sabinene 3.22% of SHJ; linalyl acetate 45.47%, linalool 7.74%, *γ*-terpineol 6.40% of SJ (Figure 2). Terpinyl acetate had anti-cholinesterase, anti-oxidation, anti-amyloidosis and neuroprotective effects, which may be used in the treatment of Alzheimer’s disease [25]. Linalyl acetate could prevent rheumatoid arthritis by antagonizing related muscle atrophy [26]. *α*-phellandrene, limonene, 1,8-cineole, *α*-terpinene and linalool are the main contributing components of ZBP odor [27]. Limonene could inhibit anxiety-related behavior [28]. Linalool and 1,8-cineole were plant-derived isoprenoids with anticancer activities in lung cancer cells with chronic nicotine exposure [29]. Myrcene could produce pine leaf flavor [30]. Although the smells of ZBP and ZBL were clearly different, they also had some similar flavor.

A total of 62 VOCs were measured, including 1 acid, 31 alkenes, 14 alcohols, 1 phenol, 2 aldehydes, 3 ketones, 9 esters and 1 alkane (Appendix A). The color indicated the logarithm of the peak area at the bottom of 10, which increased from blue to white and then to red. More VOCs of ZBP were detected by HS-SPME-GC-MS than of ZBL (Figure 3). More (1E,4E)-germacrene B, *α*-bisabolol (antiinflammatory, antibacterial) were detected in SJ. In SHJ, more 4-isopropylcyclohex-2-en-1-one, cumic alcohol, cuminaldehyde (at high concentration, strong special smell of dry tea gas, which is unpleasant; sweet smell at low concentration) and so on were detected. 1,8-cineole was detected in SHJ, SJ, SJY and SHJY, but more in SHJ and SJ. More VOCs were detected in ZBP than in its leaves, such as *γ*-terpineol (pine- and clovelike aroma), *α*-terpinene, *α*-phellandrene (black pepper aroma and sweet lotus aroma). *γ*-terpineol was heavily detected in SHJ and SJ; *γ*-terpineol had bactericidal and anticancer potential [31]. SJY’s specific VOCs were (+)-citronellal (lemon flavor, anthelmintic and antifungal). *β*-lonone, (-)-humulene epoxide II could be detected in SJY and SHJY. *α*-terpineol (clove aroma; it has strong antibacterial activity against periodontal disease and cariogenic bacteria), spathulenol, caryophyllene oxide (anti-inflammatory, anti-cancer and enhance skin permeability); all of them were detected except SJ. Terpinyl acetate (lemon and lavender fragrance), linalool (the sweet, tender and fresh flowers smell like lily of the valley), linalyl acetate (light and sweet aroma, like orange leaves, terpene-free lemon and raw pear, and like lavender flowers) were detected in all samples, and their content was higher. There was more terpinyl acetate in SHJ; more linalool and linalyl acetate in SJ, and more total ZBP than leaves.

### 3.4. Multivariate Statistical Analysis

In statistics, too many variables can increase the complexity of the analysis. PCA is an unsupervised multivariate statistical analysis method. By comparing the principal component factors, the dimensionality of the data is reduced and regularity and difference between samples are revealed [32]. Multivariate statistics of HS-GC-SPME-MS and HS-GC-IMS were obtained for model A and model B. SHJ, SJ, SHJY and SJY were clustered in the PCA of A and B, respectively. In the PCA of model A, Q2 = 0.855, indicating that the model was good. The contribution of PC1 was 72.1%, and the contribution of PC2 was 16.5%. The positive part of PC1 could well distinguish the pericarps, and the negative part of PC1 could well distinguish the leaves. In the direction of PC2, SHJ and SJ could be distinguished (Figure 4A). The PCA of model A was 0.924, indicating an excellent model fit. The contribution of PC1 was 62.7%; the contribution of PC2 was 34.7%; the distinction between SHJ and SJ was obvious; and the difference between SHJY and SJY was not obvious. The overall results were similar to HS-GC-IMS (Figure 4D). Both HS-GC-IMS and HS-SPME-GC-MS can significantly distinguish between SHJ and SJ, but the distinction between ZBL of different origins is not obvious. The reason may be due to climatic and geographical factors. Maoxian is a plateau monsoon climate with sufficient sunshine, little precipitation, a dry climate and large temperature differences; Hancheng is a continental monsoon climate with mild climate, sufficient light and more rainfall. Additionally, there were relatively fewer VOCs of ZBL, which makes the difference between SHJY and SJY appear less obvious.

OPLS-DA is a supervised method for identifying differences in samples from different categories, eliminating data that are not relevant to category information [33], here, the less characteristic components of ZBL, mostly small-molecule components detected by HS-GC-IMS. To exclude the risk of overfitting, HS-GC-IMS and HS-SPME-GC-MS were cross-validated 200 times. Q2 were less than 0.05, indicated that the results are not over-fitted (Figure 4C,F). Compounds with VIP values of >1 and *p* < 0.05 were screened by one-way ANOVA as differential markers, and components with large errors were sieved out, the differential markers marked in red in the graph (Figure 4G,H). The peak area is used as a reference to indicate the content, The content ranges from high to low corresponding to warm to cool tones. In HS-GC-IMS, 1,8-cineole for SHJ, neryl acetate, nerol, (E,E)-2,4-hexadienal, 4-terpineol, *α*-pinene, linalool oxide, acetone, *β*-ocimene, furaneol, *α*-terpinene, 2-butoxyethanol for SJ, and diethyl butanedioate for SJY were differential marker (Figure 4G). In HS-SPME-GC-MS, *α*-caryophyllene, linalool, linalyl acetate, DL-limonene for determining SJ, terpinyl acetate, *α*-thujene, piperitone, *γ*-terpineol, *α*-terpineol, sabinene for SHJ, and (+)-citronellal for CJY were differential marker (Figure 4H).

### 3.5. Comprehensive Analysis

The peak areas were used as a reference for the content of the components and were analyzed for all identified components. It was discovered that the VOC content of ZBL was significantly less than that of ZBP, and the content of SHJ was slightly less than that of SJ, regardless of HS-GC-IMS or HS-SPME-GC-MS. The results of HS-SPME-GC-MS showed that SHJ has more ketones than SJ but much less esters (Figure 5A,C). Among the VOCs detected by HS-GC-IMS, alcohols accounted for the largest proportion of both ZBL and ZBP, followed by alkenes. In addition, more abundant esters were detected in ZBP (Figure 5B). In HS-SPME-GC-MS, more ester components were detected in all samples. In SJ, the ester component was even more than 50%. The alcohol component was less and the ester component was more than in Zanthoxylum schinifolium Sieb. et Zucc [34]. Alcohols and alkenes still account for a high proportion of all samples (Figure 5D). Linalool, myrcene, 1,8-cineole, limonene and 3-nonanone were identified as the five predominant components [35]. A total of 95 components were identified by HS-GC-IMS and HS-SPME-GC-MS. α-Terpinene, 1.8-cineole, linalool, 4-terpineol, neryl acetate, acetic acid, α-pinene and β-pinene were the common components detected by those two approaches. HS-SPME-GC-MS detected more middle molecular alkenes, alcohols and esters, and HS-GC-IMS detected more small molecular alcohols, aldehydes and esters.

## 4. Conclusions

HS-GC-IMS and HS-SPME-GC-MS were used to characterize the VOCs of SHJY, SJY, SHJ and SJ rapidly, accurately, comprehensively and without contamination. The foundation was laid for the study of active ingredients. Moreover, the differences in VOCs were compared between SHJY, SHJ, SJY and SJ. Combined with multivariate statistics, thirteen and eleven differential markers were screened from HS-GC-IMS and HS-SPME-GC-MS, respectively, which contributed to the quality evaluation of ZBP and ZBL. Similarly, HS-GC-IMS and HS-SPME-GC-MS could also be applied to other VOCs of food, spices, traditional Chinese medicine and so on. Regrettably, the VOCs of ZBL are much less than ZBP. The flavor of ZBP is more acrid and strong, while the flavor of ZBL is lighter and slightly acrid, which is a valuable flavoring. Some active VOCs are present in ZBL and ZBP, and the results show the medical value of ZBP and ZPL. The different components of ZBP from different areas enriched the flavor of different regions but also led to its unstable efficacy. It is necessary to strictly control the origin and other factors to ensure that its components and properties are stable and controllable. In terms of quality control, HS-GC-IMS and HS-SPME-GC-MS will be good detection means.

## Figures and Tables

**Figure 1 foods-11-03745-f001:**
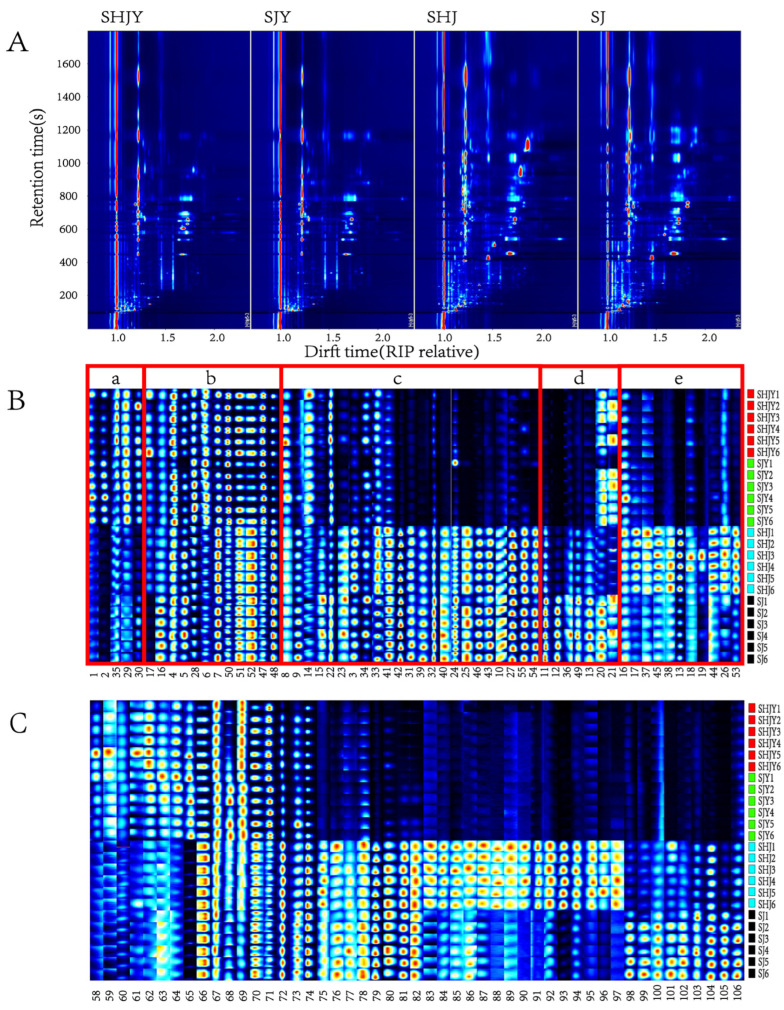
(**A**): topographic of SHJ, SJ, SHJY, SJY detected by HS-GC-IMS; (**B**,**C**): gallery plot of SHJ, SJ, SHJY, SJY detected by HS-GC-IMS. The a, b, c, d, and e are significant components of ZBL, ZBP and ZBL, ZBP, SHJ and SJ, respectively, among the identified components. (The codes of the compounds correspond to those in Table 1.)

**Figure 2 foods-11-03745-f002:**
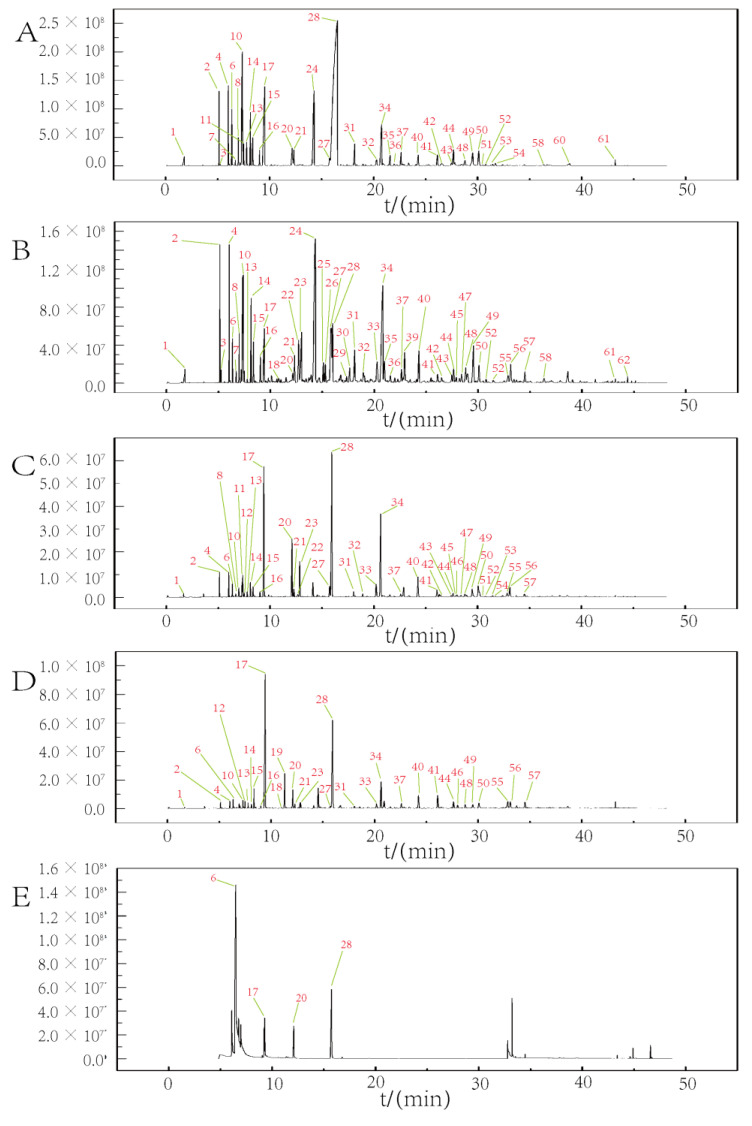
GC-MS total ion chromatogram ((**A**–**E**) corresponded to SJ, SHJ, SHJY, SJY, Reference compounds).

**Figure 3 foods-11-03745-f003:**
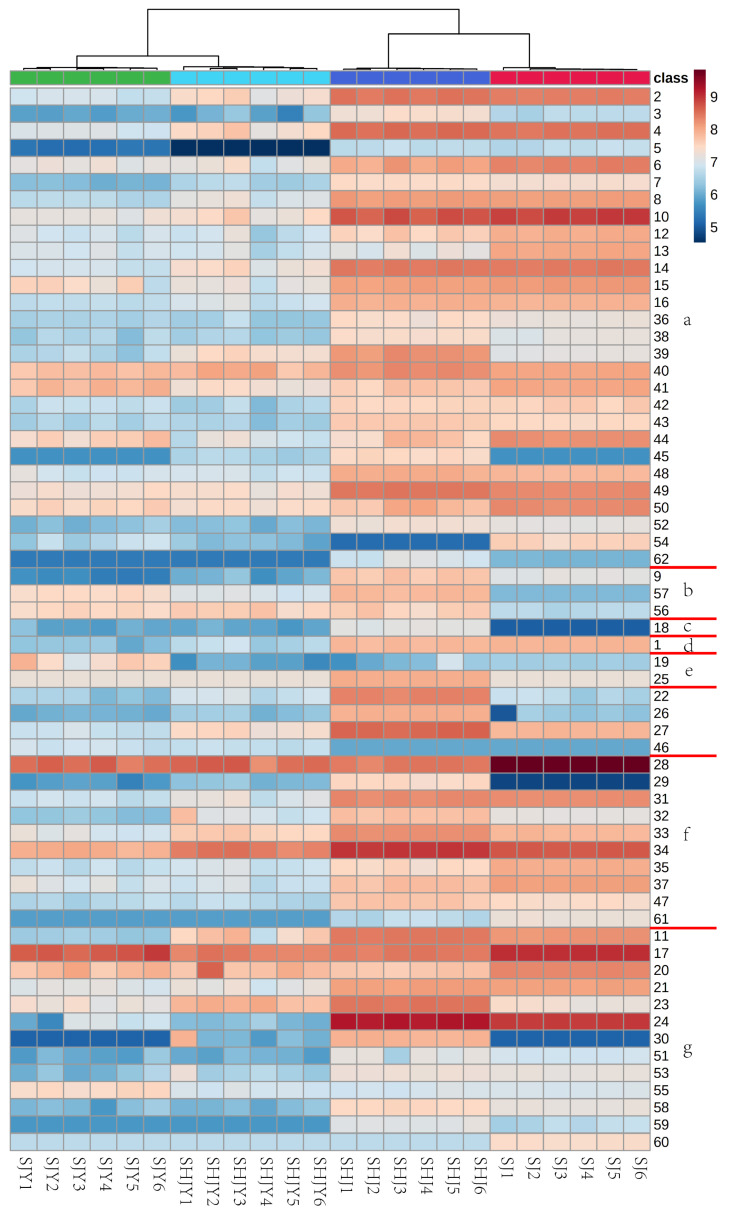
Heatmap of cluster analysis of content detected by HS-SPME-GC-MS. The a, b, c, d, e, f and g correspond to alkene, other, phenol, acid, aldehyde, ketone, ester and alcohol, respectively (the codes of the compounds correspond to those in Appendix A).

**Figure 4 foods-11-03745-f004:**
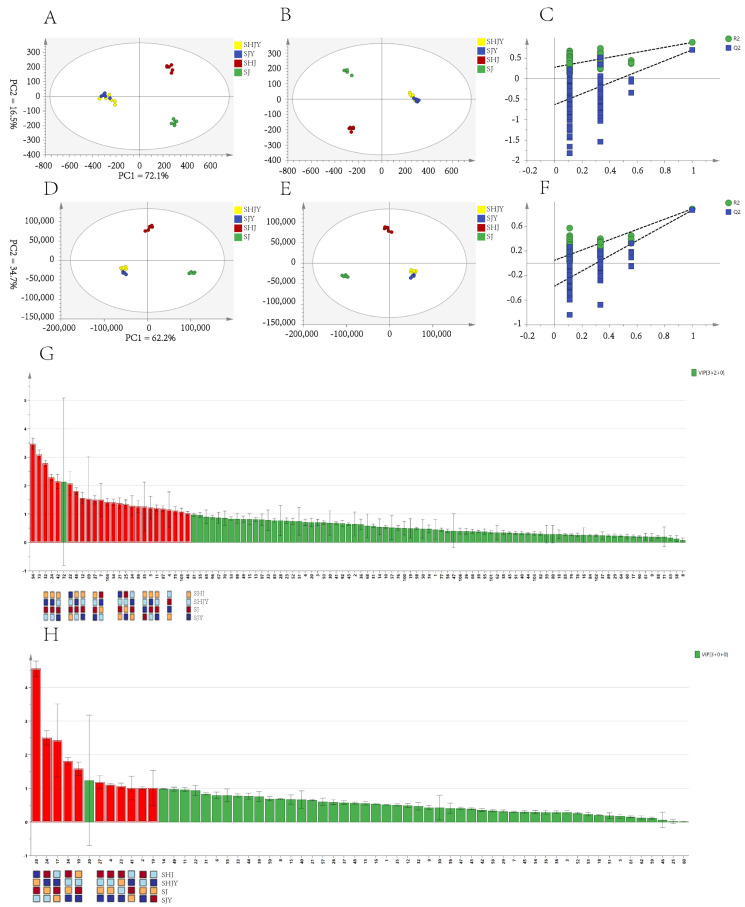
PCA, OPLS-DA, Permutation and VIP of HS-SPME-GC-MS correspond to (**A**–**C**,**G**); PCA, OPLS-DA, Permutation and VIP of HS-GC-IMS correspond to (**D**–**F**,**H**). (The codes of the compounds correspond to those in Appendix A).

**Figure 5 foods-11-03745-f005:**
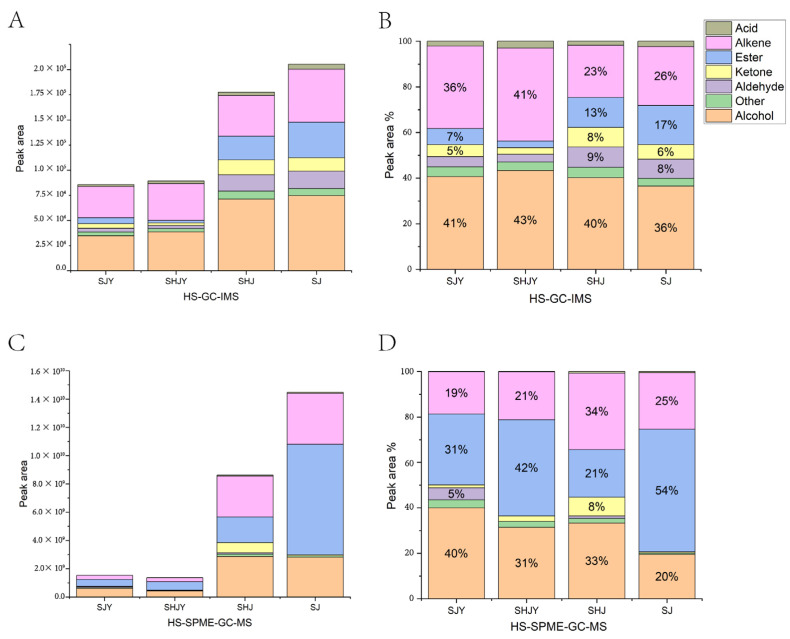
Percentage of component content of SHJ, SJ, SHJY, SJY. (**A**,**B**): the absolute and percentage contents of HS-GC-IMS; (**C**,**D**): the absolute and percentage contents of HS-SPME-GC-MS.

**Table 1 foods-11-03745-t001:** HS-GC-IMS integration parameters of VOCs in SHJ, SJ, SHJY, and SJY.

Number	Compound	CAS	Formula	RI	Rt [s]	Dt [a.u.]
1	E-2-Hexenal monomer	6728-26-3	C_6_H_10_O	840.4	338.459	1.18769
2	E-2-Hexenal dimer	6728-26-3	C_6_H_10_O	839	336.845	1.52614
3	Methyl-5-hepten-2-one	110-93-0	C_8_H_14_O	986.2	573.354	1.18323
4	*α*-Terpinene monomer	99-86-5	C_10_H_16_	1018.9	634.386	1.22448
5	*α*-Terpinene dimer	99-86-5	C_10_H_16_	1018.1	632.892	1.72976
6	1,8-Cineole monomer	470-82-6	C_10_H_18_O	1033.2	661.043	1.29673
7	1,8-Cineole dimer	470-82-6	C_10_H_18_O	1032.9	660.445	1.73863
8	2-Hexanol monomer	626-93-7	C_6_H_14_O	784.9	274.17	1.28425
9	2-Hexanol dimer	626-93-7	C_6_H_14_O	783.5	272.736	1.57117
10	2,3-Butanedione	431-03-8	C_4_H_6_O_2_	564.4	132.799	1.17274
11	Linalool oxide monomer	60047-17-8	C_10_H_18_O_2_	1078.7	746.151	1.26752
12	Linalool oxide dimer	60047-17-8	C_10_H_18_O_2_	1079.1	746.932	1.82662
13	2,3-Butanediol	513-85-9	C_4_H_10_O_2_	771.4	262.286	1.36835
14	Acetic acid monomer	64-19-7	C_2_H_4_O_2_	578.8	139.508	1.04966
15	Acetic acid dimer	64-19-7	C_2_H_4_O_2_	579.4	139.804	1.15007
16	Pentanol monomer	71-41-0	C_5_H_12_O	786	275.477	1.25575
17	Pentanol dimer	71-41-0	C_5_H_12_O	784.9	274.196	1.50928
18	2-Acetylfuran monomer	1192-62-7	C_6_H_6_O_2_	884.9	389.997	1.12002
19	2-Acetylfuran dimer	1192-62-7	C_6_H_6_O_2_	884.9	389.997	1.44292
20	*β*-Ocimene monomer	13877-91-3	C_10_H_16_	1050.1	692.666	1.25509
21	*β*-Ocimene dimer	13877-91-3	C_10_H_16_	1050	692.522	1.69652
22	4-Terpineol monomer	562-74-3	C_10_H_18_O	1168.6	914.198	1.2266
23	4-Terpineol dimer	562-74-3	C_10_H_18_O	1169.1	915.08	1.73235
24	Nerol monomer	106-25-2	C_10_H_18_O	1232.2	1033.082	1.2273
25	Nerol dimer	106-25-2	C_10_H_18_O	1228.9	1026.787	1.73416
26	Phenylacetic acid	103-82-2	C_8_H_8_O_2_	1250.7	1067.554	1.32855
27	Acetone	67-64-1	C_3_H_6_O	529.6	116.559	1.1205
28	Ethanol	64-17-5	C_2_H_6_O	509.4	107.107	1.12947
29	Ethyl acetate monomer	141-78-6	C_4_H_8_O_2_	603.3	150.951	1.09808
30	Ethyl acetate dimer	141-78-6	C_4_H_8_O_2_	601.6	150.163	1.34211
31	Furfural	98-01-1	C_5_H_4_O_2_	846.1	345.037	1.08274
32	Neryl acetate	141-12-8	C_12_H_20_O_2_	1367.5	1285.92	1.22894
33	*δ*-Octalactone	698-76-0	C_8_H_14_O_2_	1272.2	1107.751	1.28219
34	Furaneol	3658-77-3	C_6_H_8_O_3_	1083.7	755.42	1.19676
35	Diethyl butanedioate	123-25-1	C_8_H_14_O_4_	1193.7	961.041	1.29436
36	Isopropyl acetate	108-21-4	C_5_H_10_O_2_	658.7	176.819	1.16357
37	2,5-Dimethylfuran	625-86-5	C_6_H_8_O	690.4	192.347	1.35536
38	2-Methyl butanol	137-32-6	C_5_H_12_O	740.3	235.425	1.24002
39	(E)-2-Pentenal	1576-87-0	C_5_H_8_O	790	280.073	1.10915
40	Butanoic acid	107-92-6	C_4_H_8_O_2_	854.4	354.732	1.1635
41	(E,E)-2,4-Hexadienal monomer	142-83-6	C_6_H_8_O	909.9	431.414	1.11381
42	(E,E)-2,4-Hexadienal dimer	142-83-6	C_6_H_8_O	909.9	431.414	1.45388
43	Pentanoic acid	109-52-4	C_5_H_10_O_2_	898.2	409.694	1.23289
44	Dipropyl sulfide	111-47-7	C_6_H_14_S	879.3	383.546	1.15847
45	Dimethyl disulfide	624-92-0	C_2_H_6_S_2_	751.4	245.056	1.06575
46	2-Butoxyethanol	111-76-2	C_6_H_14_O_2_	898.4	410.086	1.20642
47	*α*-Pinene monomer	80-56-8	C_10_H_16_	922.1	454.074	1.22494
48	*α*-Pinene dimer	80-56-8	C_10_H_16_	922.3	454.5	1.67418
49	2-Cyclohexen-1-one	930-68-7	C_6_H_8_O	941.1	489.49	1.11358
50	*β*-Pinene monomer	127-91-3	C_10_H_16_	970.5	544.16	1.22525
51	*β*-Pinene dimer	127-91-3	C_10_H_16_	970.2	543.605	1.64775
52	*β*-Pinene tripolymer	127-91-3	C_10_H_16_	969	541.383	1.73717
53	4-Methylguaiacol	93-51-6	C_8_H_10_O_2_	1186.6	947.846	1.17166
54	Carveol	99-48-9	C_10_H_16_O	1232.9	1034.287	1.19266
55	Allyldisulfide	2179-57-9	C_6_H_10_S_2_	1066	722.333	1.19546
56	Pentanal monomer	110-62-3	C_5_H_10_O	689.4	191.444	1.18336
57	Pentanal dimer	110-62-3	C_5_H_10_O	689.9	191.877	1.43071

## Data Availability

Data is contained within the article and Appendix A.

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
