# Peer review of "The Discrimination and Characterization of Volatile Organic Compounds in Different Areas of Zanthoxylum bungeanum Pericarps and Leaves by HS-GC-IMS and HS-SPME-GC-MS"

_foods, 2022, doi:10.3390/foods11223745_

Round 1

Reviewer 1 Report

My recommendation, Major revision, is because the manuscript is very descriptive, but there needs to be a critical discussion of the results. The manuscript must compare results in volatile profile with those presented by other authors, even comparing different Citrus cultivars. 

The number of samples collected per batch must be indicated. Also, the field coordinates must be stated together with the sampling date.

 The authors must briefly describe the procedure for tentative identification by both HS-GC-IMS and HS-SPME-GC-MS. 

Was the presence of allyl disulfide confirmed in this study? Was it previously confirmed in Citrus?

 The procedure for obtaining the clean matrix and the workflow to conduct the multivariate analyses must be included in the methodology section. Was the data normalized? Was the data transformed and or scaled? This an essential information to understand the differences observed in the multivariate analysis, which can be in the methodology section.

 Are the differences observed in the Heatmap significant? The intensity of the colors doesn't indicate statistical significance; an ANOVA must be included, at least for the most abundant metabolites.

Author Response

Point 1: My recommendation, Major revision, is because the manuscript is very descriptive, but there needs to be a critical discussion of the results. The manuscript must compare results in volatile profile with those presented by other authors, even comparing different Citrus cultivars.

Response 1: We appreciate the reviewer for this suggestion. During the revision process, The results and discussion section has been optimized “

3.4. Multivariate Statistical Analysis

In statistics, too many variables can increase the complexity of the analysis. PCA is an unsupervised multivariate statistical analysis method. By comparing the princi-pal component factors, the dimensionality of the data is reduced and regularity and difference between samples are revealed [32].  Multivariate statistics of HS-GC-SPME-MS and HS-GC-IMS were obtained for model A and model B. SHJ, SJ, SHJY, and SJY were clustered in the PCA of A and B, respectively. In PCA of model A, Q2 = 0.855, indicating the model was good. the contribution of R2X[1] was 72.1% and the contribution of R2X[2] was 16.5%. the positive part of R2X[1] The positive part of R2X[1] could well distinguish the pericarps and the negative part of R2X[1] could well distinguish the leaves. in the direction of R2X[2], SHJ and SJ could be distinguished (Figure 4A). In PCA of model A was 0.924, indicating an excellent model fit. the con-tribution of R2X[1] was 62.7%; the contribution of R2X[2] was 34.7%, the distinction between SHJ and SJ was obvious, and the difference between SHJY and SJY was not obvious. The overall results were similar to HS-GC-IMS (Figure 4D). Both HS-GC-IMS and HS-SPME-GC-MS can significantly distinguish between SHJ and SJ. But the distinction between ZBL of different origins is not obvious. The reason may be due to cli-matic and geographical factors. Maoxian is a plateau monsoon climate with sufficient sunshine, little precipitation, dry climate and large temperature difference; Hancheng is a continental monsoon climate with mild climate, sufficient light and more rainfall. And the VOCs of ZBL are relatively less, which makes the difference between SHJY and SJY appear less obvious.

3.5. Comprehensive analysis

The peak areas were used as a reference for the content of the components and were analyzed for all identified components. It was discovered that the VOCs content of ZBL was significantly less than that of ZBP, and the content of SHJ was slightly less than that of SJ, regardless of HS-GC-IMS or HS-SPME-GC-MS. The results of HS-SPME-GC-MS showed that SHJ has more ketones than SJ, but much less esters (Figure 5A and Figure 5C). Among the VOCs detected by HS-GC-IMS, alcohols ac-counted for the largest proportion of both ZBL and ZBP, followed by alkenes. In addi-tion, more abundant esters were detected in ZBP (Figure 5B). In HS-SPME-GC-MS, more ester components were detected in all samples, In SJ the ester component is even more than 50 %. The alcohol component is less and the ester component is more than in Zanthoxylum schinifolium Sieb. et Zucc [34]. Alcohols and alkenes still account for a high proportion of all samples (Figure 5D). Linalool, myrcene, 1,8-cineole, limonene, 3-nonanone were identified as the five predominant components [35]. A total of 95 components were identified by HS-GC-IMS and HS-SPME-GC-MS. α-Terpinene, 1.8-cineole, linalool, 4-terpineol, neryl acetate, acetic acid, α-pinene and β-pinene were the common components detected by those two approaches. HS-SPME-GC-MS de-tected more middle molecular alkenes, alcohols and esters, and HS-GC-IMS detected more small molecular alcohols, aldehydes and esters.”

Point 2: Please revise the manuscript and correct your English writing.

Response 2: We appreciate the reviewer for this suggestion. During the revision process, English writing in the text was corrected.

Point 3: The objectives of this study are not clear. The authors should elaborate why this research is necessary and how it contribute to practical application.

Response 3: We thank the reviewer for this suggestion. The necessary for the study has been added in the citation section “Pericarps of Zanthoxylum bungeanum (ZBP) is an important seasoning in China [1]. It has a special flavor and is often used in Sichuan dishes. In addition to that it has a great demand in traditional medicine [2]. Essential oils are its main active ingredi-entsIt has anti-inflammatory [3], antiseptic [4] and other effects. Zanthoxylum bungeanum is grown in Southwest China in Sichuan, Shaanxi, Yunnan and so on [5]. Because of its wide range of cultivation, which leads to different quality. It is necessary to differentiate ZBP from different regions. Food aroma has an important influence on the evaluation results of food [6]. And the volatile terpenoids are the main source of its aroma [7]. people have different opinions about ZBP from different producing areas, its color, taste, smell and even its medicinal effects. And for the evaluation of the qual-ity of pepper, experienced operators are often required. But there is also subjectivity in judgment, affect the stability and accuracy of quality evaluation. Therefore, it is nec-essary to find a suitable approach to identify it.

Other than that, ZBL has special aroma. It is also used as a condiment and has some antioxidant [8] antibacterial activity [9]. Currently, the study of volatile organic compounds (VOCs) of ZBP is relatively abundant, however, the study of VOCs of Leaves of Zanthoxylum bungeanum (ZBL) is less. It is interesting to characterize the VOCs of ZBL and investigate the similarities and differences of their composition with ZBP.”

Point 4: In section 2, please describe the study design and steps of conducting the research. Different area of Zanthoxylum bungeanum pericarps and leaves should be described by figures.

Response 4: We appreciate the reviewer’s comment. In the revision process, added a description of the study design and steps of conducting the research. the figures (Figure S1) have upload to supplementary File.

Reviewer 2 Report

The paper presented for review is interesting, however, the discussion should be further elaborated. The authors should elaborate why this research is necessary and how it contribute to practical application. In discussion, the results of this article were compared to previous findings. It will be better if suggestion based on the data was made. The study was well-designed, time-consuming and laborious.

I have some comments to the authors as follows:

1. Please revise the manuscript and correct your English writing

2. The objectives of this study are not clear. The authors should elaborate why this research is necessary and how it contribute to practical application

3. In section 2, please describe the study design and steps of conducting the research. Different area of Zanthoxylum bungeanum pericarps and leaves should be described by figures.

Author Response

Point 1: Please revise the manuscript and correct your English writing.

Response 1: We appreciate the reviewer for this suggestion. During the revision process, English writing in the text was corrected.

Point 2: The objectives of this study are not clear. The authors should elaborate why this research is necessary and how it contribute to practical application.

Response 2: We thank the reviewer for this suggestion. The necessary for the study has been added in the citation section “Pericarps of Zanthoxylum bungeanum (ZBP) is an important seasoning in China [1]. It has a special flavor and is often used in Sichuan dishes. In addition to that it has a great demand in traditional medicine [2]. Essential oils are its main active ingredi-entsIt has anti-inflammatory [3], antiseptic [4] and other effects. Zanthoxylum bungeanum is grown in Southwest China in Sichuan, Shaanxi, Yunnan and so on [5]. Because of its wide range of cultivation, which leads to different quality. It is necessary to differentiate ZBP from different regions. Food aroma has an important influence on the evaluation results of food [6]. And the volatile terpenoids are the main source of its aroma [7]. people have different opinions about ZBP from different producing areas, its color, taste, smell and even its medicinal effects. And for the evaluation of the qual-ity of pepper, experienced operators are often required. But there is also subjectivity in judgment, affect the stability and accuracy of quality evaluation. Therefore, it is nec-essary to find a suitable approach to identify it.

Other than that, ZBL has special aroma. It is also used as a condiment and has some antioxidant [8] antibacterial activity [9]. Currently, the study of volatile organic compounds (VOCs) of ZBP is relatively abundant, however, the study of VOCs of Leaves of Zanthoxylum bungeanum (ZBL) is less. It is interesting to characterize the VOCs of ZBL and investigate the similarities and differences of their composition with ZBP.”

Point 3: In section 2, please describe the study design and steps of conducting the research. Different area of Zanthoxylum bungeanum pericarps and leaves should be described by figures.

Response 3: We appreciate the reviewer’s comment. In the revision process, added a description of the study design and steps of conducting the research. The different area of Zanthoxylum bungeanum pericarps and leaves (Figure S1) have upload to supplementary File.

Reviewer 3 Report

1. The English of the article should be checked. Especially the introduction part should be a more fluent language.

2. Chapter 2 “2.5. Why is the “Statistical Analysis” part italicized? The journal should be corrected according to the editorial boards.

3. Also 2.5. The Statistical Analysis section is very similar to another work of the authors (https://doi.org/10.1016/j.arabjc.2022.104231). Section 2.5 should be rewritten from start to finish.

4. The same situation lines 244-247 sentences are written as in the article https://doi.org/10.1016/j.arabjc.2022.104231. This situation should be noted.

5. Another case of conflict with the literature is contained in lines 301-305 sentences. https://doi.org/10.1016/j.foodchem.2021.131671

6. Especially if a new method is being developed, validation parameters such as detection limits and working range of that method should be specified. It was seen as the biggest shortcoming in the article.

Author Response

Point 1:                 The English of the article should be checked. Especially the introduction part should be a more fluent language.

Response 1: Thank the reviewers for this query. In the revision process, The introductory section has been thoroughly revised “Pericarps of Zanthoxylum bungeanum (ZBP) is an important seasoning in China [1]. It has a special flavor and is often used in Sichuan dishes. In addition to that it has a great demand in traditional medicine [2]. Essential oils are its main active ingredi-entsIt has anti-inflammatory [3], antiseptic [4] and other effects. Zanthoxylum bungeanum is grown in Southwest China in Sichuan, Shaanxi, Yunnan and so on [5]. Because of its wide range of cultivation, which leads to different quality. It is necessary to differentiate ZBP from different regions. Food aroma has an important influence on the evaluation results of food [6]. And the volatile terpenoids are the main source of its aroma [7]. people have different opinions about ZBP from different producing areas, its color, taste, smell and even its medicinal effects. And for the evaluation of the qual-ity of pepper, experienced operators are often required. But there is also subjectivity in judgment, affect the stability and accuracy of quality evaluation. Therefore, it is nec-essary to find a suitable approach to identify it.

Other than that, ZBL has special aroma. It is also used as a condiment and has some antioxidant [8] antibacterial activity [9]. Currently, the study of volatile organic compounds (VOCs) of ZBP is relatively abundant, however, the study of VOCs of Leaves of Zanthoxylum bungeanum (ZBL) is less. It is interesting to characterize the VOCs of ZBL and investigate the similarities and differences of their composition with ZBP.

GC-MS is undoubtedly a good choice for the detection of VOCs, high sensitivity, accuracy and wide detection range. But it is complex treatment of samples [10]. Solid phase microextraction (SPME) has the advantages of good extraction effect, selectivity, environmental friendliness and convenience [11]. The SPME combination with GC-MS can make up for some of its shortcomings that complex pre-processing steps. It has been reported that SPME-GC-MS combined with chemometrics could be used to ana-lyze the quality of strong aroma base liquor at different grades [12].

Ion mobility spectrometry (IMS), as a high sensitivity, fast detection, was used in the military early [13]. GC has good separation performance. By coupling GC with IMS, HS-GC-IMS had new development in food [14], cosmetics [15] and medicine [16]. In HS-GC-IMS, VOCs are formed into ions when they pass through the ionization source, and the ions drift in a weak electric field at atmospheric pressure; depending on the differences in structure, mass, charge and volume of each ion, the ions are detected at different times, providing information on the type and concentration of the analyte [17]. In addition, HS-GC-IMS also has the advantages of high sensitivity and environ-mental friendliness and no sample pretreatment [18]. Previously it has been reported that HS-GC-IMS can identify Pericarpium Citri Reticulatae and its counterfeits [19]. GC-MS tends to be qualitative and quantitative. HS-GC-IMS tends to identify samples, and can also detect small odor molecules which are not detected by GC-MS. GC-MS and HS-GC-IMS each have their own advantages. The combination of these two ap-proaches can get better evaluation results [20]. It has reported that the Liuyang Douchi was determined by HS-GC-IMS and HS-SPME-GC-MS [21].

 Maoxian and Hancheng are historical pepper production areas in China. Maoxian pepper has been appraised as China's national geographical indication products. The purpose of this study was to characterize the VOCs of ZBP of Hancheng (SHJ), ZBL of Hancheng (SHJY), ZBP of Maoxian (SJ) and ZBL of Maoxian (SJY) based on HS-GC-IMS and HS-SPME-GC-MS and investigate its differential components. Laying the foundation for the study of its active ingredients. The fingerprint of HS-GC-IMS data and the heat map of HS-SPME-GC-MS data was established. The PCA and OPLS-DA were used to distinguish the samples and find the difference VOCs between them. Differential marker screening of ZBP and ZBL by One-way ANOVA and variable importance in projection (VIP) evaluation for the differential marker”.

Point 2: Chapter 2 “2.5. Why is the “Statistical Analysis” part italicized? The journal should be corrected according to the editorial boards.

Response 2: Thank the reviewers for suggestion. In the revised version, the font problem has been corrected.

Point 3: Also 2.5. The Statistical Analysis section is very similar to another work of the authors (https://doi.org/10.1016/j.arabjc.2022.104231). Section 2.5 should be rewritten from start to finish.

Response 3: We thank the reviewers for comments. In the revised version, Section 2.5 has rewritten “Analysis of HS-GC-IMS data based on Vocal, drift time and RI were used as crite-ria for component identification. Analysis of GC-MS data based on Masshunter. NIST17 (match > 80, RI) as well as standard compounds were used for identification analysis. Multivariate statistical analysis by simca 14.1 (Umetrics,Sweden). Data were normalized by peak area and normalization and scaled using pareto scaling. Plotting heat maps on MetaboAnalyst 5.0 (https://www.metaboanalyst.ca/). Compound odor and partial activity query from ChemicalBook (www.chemicalbook.com).”

Point 4: The same situation lines 244-247 sentences are written as in the article https://doi.org/10.1016/j.arabjc.2022.104231. This situation should be noted.

Response 4: Thanks to the reviewers' suggestions, we have revised the description of this section “In statistics, too many variables can increase the complexity of the analysis. PCA is an unsupervised multivariate statistical analysis method. By comparing the princi-pal component factors, the dimensionality of the data is reduced and regularity and difference between samples are revealed.”

Point 5: Another case of conflict with the literature is contained in lines 301-305 sentences. https://doi.org/10.1016/j.foodchem.2021.131671.

Response 5: Thank you for your suggestion, duplicate content has been changed in revision.” HS-GC-IMS and HS-SPME-GC-MS were used to characterize the VOCs of SHJY, SJY, SHJ, SJ rapidly, accurately, comprehensively and without contamination. Laid the foundation for the study of active ingredients. Moreover, Compared the differences in VOCs between SHJY, SHJ, SJY and SJ. Combined with multivariate statistics, Thirteen and eleven differential markers were screened from HS-GC-IMS and HS-SPME-GC-MS, respectively. It contributed to the quality evaluation of ZBP and ZBL. Similarly, HS-GC-IMS and HS-SPME-GC-MS could also be applied to other VOCs of food, spices, traditional Chinese medicine and so on. Regrettably, the VOCs of ZBL are much less than ZBP. The flavor of ZBP is more acrid and strong, while the flavor of ZBL is lighter and slightly acrid. Which are a valuable flavoring. Some active VOCs are present in ZBL and ZBP. The side shows the medical value of ZBP and ZPL. The differ-ent components of ZBP from different areas enriched the flavor of different regions. But it also led to the instable of its efficacy. It is necessary to strictly control the origin and other factors for its components and properties are stable and controllable. In terms of quality control, HS-GC-IMS and HS-SPME-GC-MS will be a good detection means.”

Point 6: Especially if a new method is being developed, validation parameters such as detection limits and working range of that method should be specified. It was seen as the biggest shortcoming in the article.

Response 6: Thank the reviewers for suggestion. For the validation parameters, we weighed an equal amount of sample as QC, Each take 6 QC measured by HS-GC-IMS and HS-SPME-GC-MS, and selected 9 and 6 major components to calcu-late the relative standard deviation to verify the reproducibility, RSD < 10.47% and 6.67%, respectively (Table S2).

Round 2

Reviewer 1 Report

In the reviewer's opinion, the revised version of the manuscript has reached sufficient quality to be published in Foods.

Reviewer 3 Report

Dear Editor;

Last version of the article is acceptable.